# The Analysis of the Power Law Feature in Complex Networks

**DOI:** 10.3390/e24111561

**Published:** 2022-10-29

**Authors:** Xiaojun Zhang, Zheng He, Liwei Zhang, Lez Rayman-Bacchus, Shuhui Shen, Yue Xiao

**Affiliations:** 1School of Mathematical Sciences, University of Electronic Science and Technology of China, Chengdu 611731, China; 2School of Management and Economics, University of Electronic Science and Technology of China, Chengdu 611731, China; 3School of Mathematical Sciences, Dalian University of Technology, Dalian 116024, China; 4Business School, University of Winchester, Winchester SO22 4NR, UK

**Keywords:** power law, preferential attachment, random addition and deletion, complex networks

## Abstract

Consensus about the universality of the power law feature in complex networks is experiencing widespread challenges. In this paper, we propose a generic theoretical framework in order to examine the power law property. First, we study a class of birth-and-death networks that are more common than BA networks in the real world, and then we calculate their degree distributions; the results show that the tails of their degree distributions exhibit a distinct power law feature. Second, we suggest that in the real world two important factors—network size and node disappearance probability—will affect the analysis of power law characteristics in observation networks. Finally, we suggest that an effective way of detecting the power law property is to observe the asymptotic (limiting) behavior of the degree distribution within its effective intervals.

## 1. Introduction

In 1999, Barabási and Albert published the seminal article “Emergence of scaling in random networks” in *Science* [1], in which they suggested that growth and preferential attachment are two key characteristics for real-world networks, and further suggested that networks with the power law feature widely exist in the real world. Over the last 20 years, their contribution has exerted a significant impact on network science research [2,3,4,5].

Generally, a network with the power law feature means that the tail of its degree distribution follows a power law [6,7,8,9,10,11]. In other words, there is a positive integer $k^{\prime }$, and when $k>k^{\prime }$, the probability of a node with degree k is:
$$P\left(k\right)\propto k^{\alpha }$$
where α is the exponent of the power law.

For a long time now, researchers and practitioners have believed that the power law feature is common in real-world networks, such as the Internet, scientific co-authoring networks, metabolic networks, and biological networks [12,13,14,15,16,17,18,19,20,21], and numerous results are based on this property (with more than 35,000 citations by Google Scholar). However, more recently, some have begun to question its universality [11,22,23,24,25,26]. For example, Tanaka [22] argues that the metabolite degree distributions at the module level follow an exponential distribution. Similarly, Lima-Mendez and Helden [25] found that the degree distributions in many biological networks are not subject to a power law. More broadly, Stumpf and Porter [11] argue that “most reported power laws lack statistical support and mechanistic backing”. In addition, Broido and Clauset [26], employing statistical tools, analyzed nearly 1000 networks in the social, biological, technological, and informational domains, concluding that scale-free networks are empirically rare. These contradictory and critical studies have shaken the cornerstone of complex network theory, provoking widespread controversy.

Such empirical challenges have spurred researchers to employ alternative or additional criteria to describe the degree distribution, such as low degree saturation, high degree cutoffs, and improved goodness of fit [7,8,21,27,28]. These refinements are thought to provide a better understanding of deviation from the pure power law. Still, some continue to doubt the reliability or validity of empirical data [9,29,30].

Clearly, real-world networks are dynamic—a feature that must be captured in any attempt to explain power law behavior. We suggest that a critical method for informing this debate is to first establish a generic, theoretical, and realistic evolutionary mechanism [11,21,26] in order to examine the power law feature. This theoretical mechanism should include four steps: theoretical model building, degree distribution solving, power law feature judgment, and interpretation of empirical results.

In this paper, we first study a class of birth-and-death networks that are more common than BA networks in the real world, and then we calculate their degree distributions. Our results show that the tails of their degree distributions exhibit a distinct power law feature, providing robust theoretical support for the ubiquity of the power law feature. Second, we suggest that in the real world two important factors—network size and node disappearance probability—point to the existence of the power law feature in the observed networks. As network size reduces, or as the probability of node disappearance increases, the power law feature becomes increasingly difficult to observe. Finally, we suggest that an effective way of detecting the power law property is to observe the asymptotic (limiting) behavior of the degree distribution within its effective intervals.

## 2. The Power Law Feature Analysis of Complex Networks

### 2.1. Model

For any network, there are two basic elements: nodes, and edges connecting different nodes. In numerous real-world networks (e.g., social, ecological, business, and biological networks), nodes are agents with life cycles and may possess intelligence. During the evolving processes of these networks, nodes may enter or exit randomly, reflecting the birth and death of network nodes [31,32,33,34]. Meanwhile, these agents (nodes) may exhibit differing capabilities, making them unequal in terms of resources and positions in their networks. Those with scarce resources or occupying critical positions will be more attractive, and new entrants will prefer to establish linkages with nodes that provide what they need. These behaviors make preferential attachment [35,36,37,38,39] widespread in real-world networks, such as the “rich-get-richer” phenomenon [40,41,42,43]. Accordingly, we suggest that random addition/deletion of nodes and preferential attachment are two universal behaviors in the evolution of real-world networks.

Based on the above, we established a network model characterized by random addition and deletion of nodes as well as preferential attachment. The evolving rules are as follows:

(1)The initial network is a complete graph with m+1 (m≥1) nodes;(2)At each unit of time, randomly delete a node from the network with probability q (0≤q<1/2), or add a new node to the network with probability p=1−q and connect it with m old nodes of the network by preferential connection. That is, the probability that the new node connects with an old node *i* depends on the degree $k_{i}$ of node *i*, i.e., $\pi_{i}=\frac{mk_{i}}{\sum_{j}k_{j}}$.

*Note*:

(a)Considering the real world, any network size has its lower bound $n_{0}$. Here, we assume that $n_{0}=1$. This assumption will not affect the power law feature of the model.(b)If at time t, a node is deleted, then all the edges incident to the removed node are also removed from the network; thus, the degree of its neighbors decreases by one.(c)If at time t, a new node is added to the network and the network size is less than m, then the new node is connected with all old nodes.

In the case of q=0, this model equates to the BA model. Thus, the BA model is a special case of our model. 

### 2.2. Steady-State Degree Distribution

Employing the stochastic process rules (SPR) method [44], we obtained the equation of (steady-state) degree distribution *K* for differing *k* (see Appendix A for details).
(1)$$\left(p+\frac{k}{2}+kq\right)P\left(k\right)=\left\{ \begin{array}{l} qP\left(1\right)\;\;\;\;\;\;\;\;\;\;\;\;\;\;\;\;k=0\\ \left(m+1\right)qP\left(m+1\right)+\frac{m-1}{2}P\left(m\;-\;1\right)+p\;\;\;\;\;k=m\\ \left(k+1\right)qP\left(k+1\right)+\frac{k\;-\;1}{2}P\left(k-1\right)\;\;\;\;\;1\leq k,\;k\neq m \end{array}\right.$$
where P(k)=P{K=k}. When q=0, the equations of steady-state degree distribution *K* are as follows:(2)$$\left\{ \begin{array}{l} \left(m+2\right)P\left(m\right)=2\\ \left(r+2\right)P\left(r\right)=\left(r-1\right)P\left(r-1\right)\;\;r\geq m+1 \end{array}\right.$$
Equation (2) also represents the BA model.

By using the probability-generating function, the solutions to the above equations can be obtained First, we need to normalize Equation (1) and then calculate the normalized equations using the probability-generating function. Here, let
$$\Pi \left(k\right)=P\left(k\right)+\beta_{k}\;\;k=0,1,2,\cdots$$
where $\beta_{k}=0\;\left(k\geq m\right)$, and for 0≤k<m, $\beta_{k}$ satisfies the following linear equations:Aβ=η
where:$$A=\left[\begin{array}{ccccccc} -a_{0} & b_{1} & 0 & & & & \\ & c_{1} & -a_{2} & b_{3} & & & \\ & & c_{2} & -a_{3} & b_{4} & & \\ & & & \ddots & \ddots & \ddots & \\ & & & & c_{m-3} & -a_{m-2} & b_{m-1}\\ & & & & & c_{m-2} & -a_{m-1}\\ & & & & & & c_{m-1} \end{array}\right],\;\beta =\left[\begin{array}{c} \beta_{0}\\ \beta_{1}\\ \beta_{2}\\ \vdots \\ \beta_{m-3}\\ \beta_{m-2}\\ \beta_{m-1} \end{array}\right],\;\eta ={\left[\begin{array}{c} 0\\ 0\\ 0\\ \vdots \\ 0\\ 0\\ p \end{array}\right]}_{m\times 1}$$
$$a_{i}=p+\frac{i}{2}+iq,\;b_{i}=iq,\;c_{i}=\frac{i}{2}\;\;i=0,1,2,3,\cdots$$
so:$$\beta \;=\;A^{-1}\eta$$
Then, Equation (1) can be written as follows:(3)$$\left\{ \begin{array}{l} p\Pi \left(0\right)=q\Pi \left(1\right)\\ \frac{3}{2}\Pi \left(1\right)=2q\Pi \left(2\right)+\hat{p}\\ \left[2+q\right]\Pi \left(2\right)=3q\Pi \left(3\right)+\frac{1}{2}\Pi \left(1\right)\\ \;\;\;\;\vdots \\ \left(p+\frac{r}{2}+rq\right)\Pi \left(r\right)=\left(r+1\right)q\Pi \left(r+1\right)+\frac{r-1}{2}\Pi \left(r-1\right)\\ \;\;\;\;\vdots \end{array}\right.$$
where $\hat{p}=\frac{3}{2}\beta_{1}-2q\beta_{2}$.

To solve Equation (3), we can use the probability-generating function. Let
(4)$$G\left(x\right)=\sum_{k=0}^{\infty }\Pi \left(k\right)x^{k}$$

Multiplying $x^{k}$ to both sides of the (*k +* 1)th equation in Equation (3), and adding all of them together, we can get
(5)$$2pG\left(x\right)=G^{\prime }\left(x\right)\left(x^{2}-\left(1-2q\right)x+2q\right)+2\hat{p}x$$
Solving Equation (5), we can get
(6)$$G\left(x\right)={\left(\frac{1-x}{2q-x}\right)}^{\frac{2p}{1-2q}}\int_{x}^{2q}\frac{2\hat{p}t}{\left(1-t\right)\left(2q-t\right)}{\left(\frac{2q-t}{1-t}\right)}^{\frac{2p}{1-2q}}dt$$
and we may have
(7)$$\begin{array}{l} G\left(x\right)=\;\frac{2\hat{p}q}{p}-2\hat{p}\sum_{i=0}^{\infty }\frac{1}{\frac{2p}{1-2q}+i+1}{\left(\frac{2q-x}{1-x}\right)}^{i+1}\\ \;\;\;\;\;=\frac{2\hat{p}q}{p}-2\hat{p}\sum_{i=0}^{\infty }\frac{1}{\frac{2p}{1-2q}+i+1}{\left(1+\frac{2q-1}{1-x}\right)}^{i+1} \end{array}$$
Employing Taylor expansion for Equation (7), we can get
(8)$$\begin{array}{l} G\left(x\right)=\frac{2\hat{p}q}{p}-2\hat{p}\sum_{i=0}^{+\infty }\frac{{\left(2q\right)}^{i+1}}{\frac{2p}{1-2q}+i+1}+\\ \;\;\;\;\;\;\;\;\;\;\;2\hat{p}\sum_{r=1}^{+\infty }\left[\sum_{i=1}^{+\infty }\frac{1}{\frac{2p}{1-2q}+i}\sum_{j=1}^{i}{\left(-1\right)}^{j+1}{\left(1-2q\right)}^{j}C_{i}^{j}C_{j+r-1}^{r}\right]x^{r} \end{array}$$

Comparing with Equation (4), Π(k) can be obtained as follows:(9)$$\Pi \left(k\right)=\left\{ \begin{array}{l} \frac{2\hat{p}q}{p}-2\hat{p}\sum_{i=0}^{+\infty }\frac{{\left(2q\right)}^{i+1}}{\frac{2p}{1-2q}+i+1}\;\;\;\;\;\;\;\;\;\;k=0\\ 2\hat{p}\sum_{i=1}^{+\infty }\frac{1}{\frac{2p}{1-2q}+i}\sum_{j=1}^{i}{\left(-1\right)}^{j+1}{\left(1-2q\right)}^{j}C_{i}^{j}C_{j+k-1}^{k}\;\;k\geq 1 \end{array}\right.$$

Then, the solution of Equation (1) is as follows:(10)$$P\left(k\right)=\left\{ \begin{array}{l} \frac{2\hat{p}q}{p}-2\hat{p}\sum_{i=0}^{+\infty }\frac{{\left(2q\right)}^{i+1}}{\frac{2p}{1-2q}+i+1}-\beta_{0}\;\;\;\;\;\;\;\;\;\;\;\;k=0\\ 2\hat{p}\left[\sum_{i=1}^{+\infty }\frac{1}{\frac{2p}{1-2q}+i}\sum_{j=1}^{i}{\left(-1\right)}^{j+1}{\left(1-2q\right)}^{j}C_{i}^{j}C_{j+k-1}^{k}\right]-\beta_{r}\;\;\;\;1\leq k\leq m-1\\ 2\hat{p}\left[\sum_{i=1}^{+\infty }\frac{1}{\frac{2p}{1-2q}+i}\sum_{j=1}^{i}{\left(-1\right)}^{j+1}{\left(1-2q\right)}^{j}C_{i}^{j}C_{j+k-1}^{k}\right]\;\;\;\;\;k\geq m \end{array}\right.$$

In order to closely observe the property of its tails, Figure 1 illustrates the solution for m=4 with different values of q.

It is easy to observe that when k>200 (Figure 1A) and k>3000 (Figure 1B), the tails approximate to straight lines, implying that the degree distributions of the networks exhibit distinct power law tails. Similar results can be observed with other values of m and q. Therefore, the proposed birth-and-death network model shows the power law feature, providing theoretically robust support for the ubiquity of the power law feature in real-world networks.

### 2.3. Power Exponent

Meanwhile, this study can also improve our understanding of the power exponent α. According to our model, for sufficiently large k, we have $P\left(k\right)\propto k^{\alpha }$, where α can be obtained directly as follows: From Equation (1), for sufficiently large *k*, we can get
(11)(2p+k+2kq)P(k)=2(k+1)qP(k+1)+(k−1)P(k−1)
Noticing that P(k)≠0, then
(12)$$\frac{2q\left[kP\left(k\right)-\left(k+1\right)P\left(k+1\right)\right]}{P\left(k\right)}=\frac{\left(k-1\right)P\left(k-1\right)-kP\left(k\right)}{P\left(k\right)}-2p$$

Let
(13)$$P\left(k\right)\propto \lambda k^{\alpha }$$
Taking Equation (13) into Equation (12), and when k→+∞, we can obtain
(14)$$\underset{k\to +\infty }{\lim}\frac{2q\left[k^{\alpha +1}-{\left(k+1\right)}^{\alpha +1}\right]}{k^{\alpha }}=\underset{k\to +\infty }{\lim}\frac{{\left(k-1\right)}^{\alpha +1}-k^{\alpha +1}}{k^{\alpha }}-2p$$
That is:(15)2q(α+1)=α+1+2p
Hence:(16)$$\alpha =-\frac{3-4q}{1-2q}$$

Since α is monotonically decreasing with q, it is easy to see that α≤−3 for all $0\leq q<\frac{1}{2}$. In particular, if and only if q=0, we have α=−3. 

As illustrated in Figure 2, α will change with the node disappearance probability *q*, explaining why various power law networks have different exponents in the real world. In particular, when q=0, our evolving model degenerates into the BA model with α=−3. Moreover, we may find that α changes very slowly from −3 to −7 as *q* increases from 0 to 0.4, but drops sharply once q>0.4. In contrast with studies that highlight differing values of α resulting from linkage changes or aging [7,8], our findings emphasize the significant impact of *q* on α and highlight their monotonic (decreasing) relationship.

## 3. The Analysis of Realistic Results

As we know, the power law describes the statistical characteristics of nodes with large degrees. In general, since the probability of these nodes appearing in a complex network is relatively very small, a tiny sampling error may significantly affect their sampling frequencies, leading to a misjudgment of the power law feature. Our theoretical results further show that as the network size *n* decreases or the node disappearance probability *q* increases, the power law property becomes increasingly difficult to observe in real-world networks.

For any empirical study, network data are critical for observing the power law feature of a network. Generally speaking, there are two types of data: whole-network data, and sampling data. When the empirical data are whole-network data, the network size *n* will affect the deviation of its degree distribution tail from the power law tail. Indeed, the power law tail is obtained as n→+∞, meaning that the smaller the network size *n*, the larger the deviation. As Figure 3 shows, for n=100 and 500, the tails of the degree distributions deviate greatly from the power law. However, with the increase in *n*, the tails of the degree distributions show an asymptotic behavior; that is, the range of *k* subjected to the power law feature gradually becomes wider with the growth of *n*. From Figure 3, we can see that as *n* grows from 2000 to 10,000, the range of *k* that follows the power law property also increases from 10≤k≤100 to 10≤k≤200. Compared with the high degree cutoffs [21,22,23,24,25,26,27,28], this asymptotic behavior provides a dynamic lens for observing the power law characteristics of real-world networks.

When the empirical data are obtained by sampling, in addition to the impact of network size n, the node disappearance probability *q* will also affect the effective interval for detecting power law characteristics. Commonly, in these empirical studies, frequency $f_{k}$ is used as a proxy for P(k), and for any sampling data there is a sampling error Δ, satisfying $\left|f_{k}-P\left(k\right)\right|\leq \Delta$, i.e., $0\leq f_{k}\leq \Delta +P\left(k\right)$. As $\underset{k\to +\infty }{\lim}P\left(k\right)=0$, for a fixed *q*, there exists a specific degree $k_{q}$*,* and for any $k>k_{q}$, P(k)≤0.1Δ. Thus, for $k>k_{q}$, we have $0\leq f_{k}\leq \Delta +P\left(k\right)\approx \Delta$, implying that $f_{k}$ cannot be employed as a proxy for P(k). Therefore, the effective interval to observe the power law feature is $\left[m,\;k_{q}\right]$. Moreover, with the increase in *q*, $k_{q}$ gradually decreases, and the effective interval $\left[m,\;k_{q}\right]$ will be narrower, making the discernment of the power law feature more difficult.

Figure 4 shows the changes in $k_{q}$ with *q* under different sampling errors. Taking sampling error $\Delta =10^{-7}$ [1,9,14,27] as an example, for q=0, we have $k_{q}=1587$, meaning that the effective interval for detecting the power law feature is [4,1587]. Furthermore, when q≥0.4, we have $k_{q}\leq 182$, which contradicts the results of Figure 1, where the power law tails are observed only for k>200. Thus we suggest that for q≥0.4, the power law feature is imperceptible, i.e., almost impossible to perceive in an empirical study. It should be noted that for q=0.4, we have α=−7, showing that α is between −7 and −3 for the observed networks. This theoretical result is consistent with empirical findings of α∈[−8,−2] in [14,15,16,17,18,19,20,21], which also shows the reliability and validity of our proposed model. Combining the influences of both *n* and *q*, we suggest that a practical way to detect the power law property of a real-world network is to observe the asymptotic behavior of the degree distribution within its effective interval $\left[m,\;k_{q}\right]$.

## 4. Conclusions

We conclude that the power law feature is proven to be universal in theory but difficult to observe in reality. Complex networks in the real world exhibit diverse evolving mechanisms [45,46,47,48,49,50]. Although random addition and deletion of nodes and preferential attachment were used to establish our birth-and-death network model, it is necessary to investigate other evolving rules and examine their limit properties [48,49,50]. Such further studies may help enrich our understanding of the power law mechanism, as well as revealing more nuanced features.

## Figures and Tables

**Figure 1 entropy-24-01561-f001:**
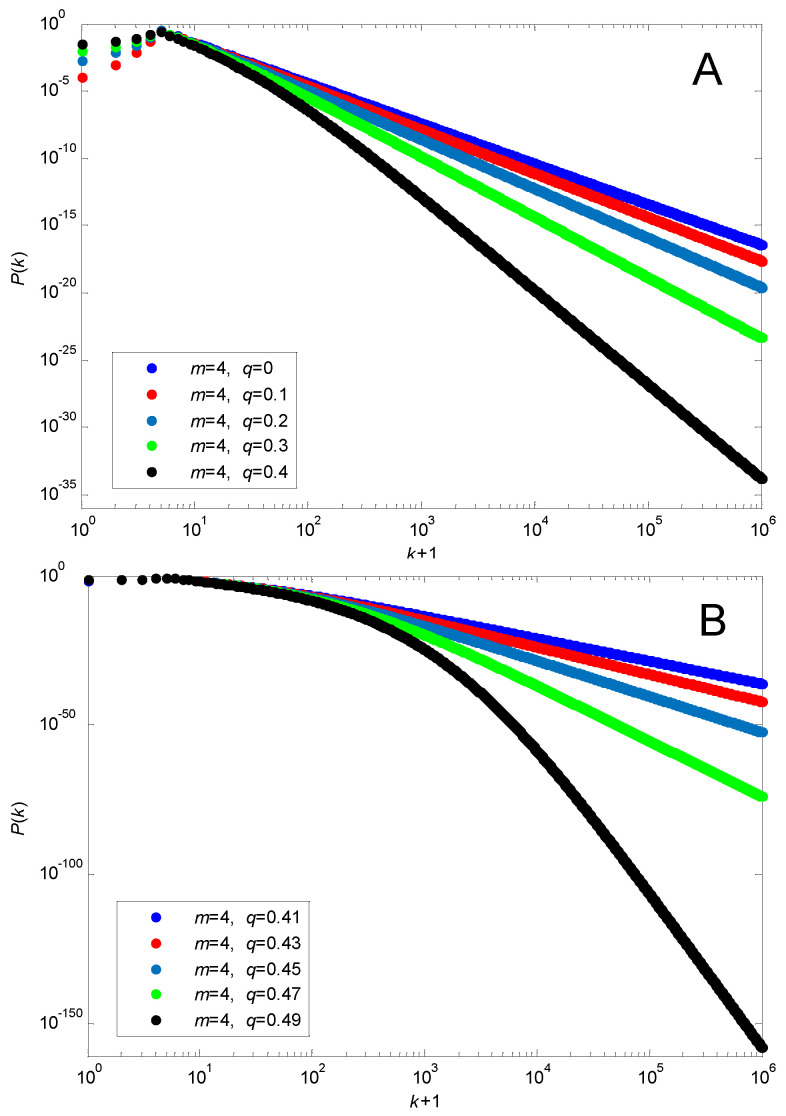
The steady-state degree distribution for m=4. (**A**) 0≤q≤0.4; (**B**) 0.4<q<0.5.

**Figure 2 entropy-24-01561-f002:**
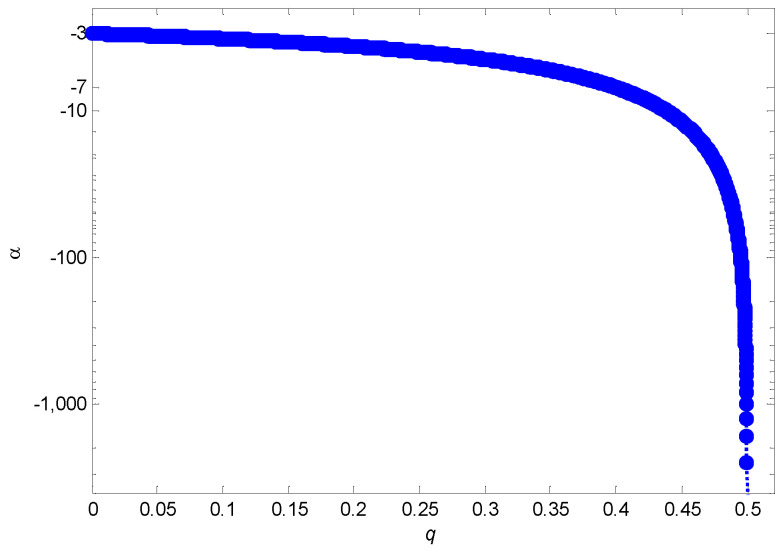
Relationship between the exponent α and the node disappearance probability *q*.

**Figure 3 entropy-24-01561-f003:**
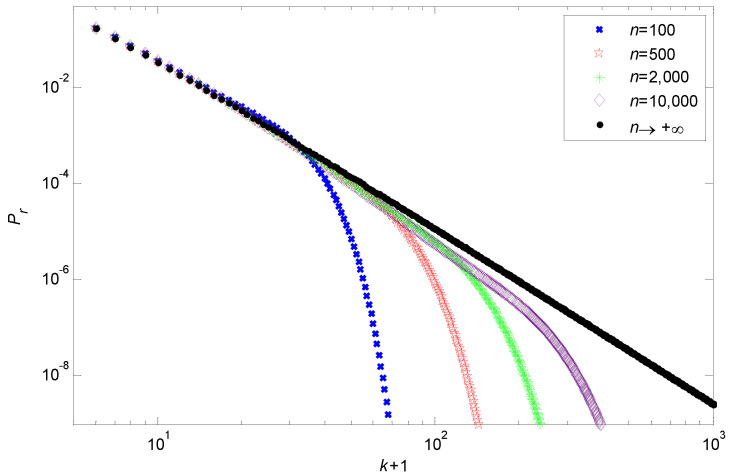
The degree distributions for different network size *n* under m=4 and q=0.2.

**Figure 4 entropy-24-01561-f004:**
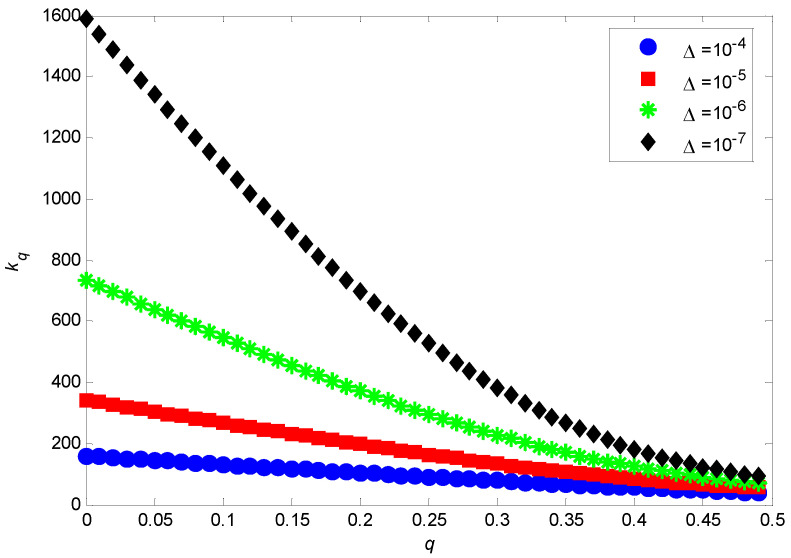
Relationship between $k_{q}$ and q for different values of Δ (*m* = 4).

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
