# Peer review of "The Analysis of the Power Law Feature in Complex Networks"

_entropy, 2022, doi:10.3390/e24111561_

Round 1
Reviewer 1 Report
Authors investigate a model with random preferential attachment and removal of nodes and found that it can generate power law degree distributions with various exponents. I found the paper interesting, clear and well written. The results are sound. I have found no weak points. The only concerns is about the gmail address of Lez Rayman-Bacchus: why is he not signing with an official address?
Author Response
Dear Reviewer,
Sincerely thank you for your to-the-point comments and suggestions.Your comments are in red while our answers are in black.
The only concerns is about the gmail address of Lez Rayman-Bacchus: why is he not signing with an official address?
Professor Lez Rayman Bacchus is a British scholar who has worked with us for a long time. He is mainly responsible for polishing the English language and preparing the final paper. He has been using Gmail to contact us. We respect his choice.
Reviewer 2 Report
In this well-written manuscript, the authors consider the birth-and-death network, an extension of the famous BA network, and calculate its degree distribution analytically. The computation is complicated, but the results appear to be reasonable. They also mention the correspondence between theoretical calculations and actual empirical networks. Although it may seem a bit out of date as a research method, I think it is interesting to the readership of Entropy, and would recommend it for acceptance after the minor points as follows.
In my opinion, Eq. (1) is one of the most important results, but its derivation is not very well explained. In Appendix A, there is a big leap just before Eq. (1.17), and I could not follow this part. In addition, I could not follow the expansion from Eq. (6) to Eq. (7). An additional explanation of the expansion of these equations would be nice.
Line 214: The power exponent α of the empirical network is stated to be between -8 and -2, but in many textbooks, it is stated to be between -3 and -2. It would be good to have some examples with power exponents of -8 to -3.
Can you think of an intuitive reason why the area of the power distribution becomes narrower (i.e. k_q increases) as q increases?
Minor
Line 214: I think the reference number 24 is a mistake for 44.
Eq. (1.16): I feel that it is better to specify that it is an infinite limit of t, considering the effect that when a node is deleted, its links are also reduced.
Reviewer 3 Report
In their article “The Analysis of Power Law Feature in Complex Networks”, Xiaojun Zhang, Zheng He, Liwei Zhang, Lez Rayman-Bacchus, Shuhui Shen and Yue Xiao propose a generic theoretical framework in order to examine the power law property. Therefore, the authors demonstrate that network size and node disappearance probability are crucial to observe power law characteristics of a given networks.
The study presented by the authors is of high quality and very interesting and intriguing. In particular, it is very fascinating that the results show the power law characteristics to be universal in theory but difficult to observe in reality. The work of the authors makes this point clear on the one hand using simulation and on the other by analyzing real word data. Regarding their simulation they use specific node and edge probabilities that might have an impact on the results but are scientifically sound in their rationale design choice. Therefore, it might be interesting, however not necessary to use different random networks approaches for simulation.
Author Response
Dear Reviewer,
Sincerely thank you for your to-the-point comments and suggestions. They are very helpful. Bellow we reply to each of your comments. Your comments are in red while our answers are in black.
Therefore, it might be interesting, however not necessary to use different random networks approaches for simulation.
We agree with you, because the statistical characteristics of complex networks are studied in an average sense.